# Parenting Stress and Its Influencing Factors Among Chinese Parents in Parent–Grandparent Co-Parenting Families: A Latent Profile Analysis

**DOI:** 10.3390/bs15040533

**Published:** 2025-04-15

**Authors:** Tongyao Wang, Hongyan Cheng

**Affiliations:** School of Education, Central China Normal University, Wuhan 430079, China; wty@mails.ccnu.edu.cn

**Keywords:** parenting stress, parent–grandparent co-parenting, latent profile analysis, preschool children

## Abstract

Guided by family systems theory and the parenting process model, this study aimed to identify distinct profiles of parenting stress and examine their associations with parental characteristics, social–contextual factors, and child factors. A sample of 303 parents of preschool children (52.5% boys, mean child age = 61.17 months) from six urban kindergartens in southern and northern China participated in this study. Latent profile analysis (LPA) identified four distinct parenting stress profiles: the low parenting stress profile (12.9%), middle parenting stress profile (39.3%), high parenting stress profile (40.6%), and very high parenting stress profile (7.2%). Multinomial logistic regression analysis revealed that these profiles were significantly associated with parenting self-efficacy, the parent–grandparent co-parenting relationship, the co-parenting structure, family income, and the child’s gender. These findings advance our understanding of the heterogeneity of parenting stress within Chinese parent–grandparent co-parenting families and offer theoretical and practical implications for future research and the development of targeted family support interventions.

## 1. Introduction

The phenomenon of intergenerational co-parenting, where grandparents actively participate in raising grandchildren alongside parents, has become a global trend ([56]). Driven by demographic change and shifting work–family demands, the proportion of Dutch grandparents providing childcare rose from 23 percent in 1992 to 66 percent in 2015 ([20]), while more than 7.3 million U.S. grandparents lived with their grandchildren in 2015 and over one-third served as primary caregivers ([56]). Grandparental involvement is long-standing in many Asian societies ([30]) and is especially pronounced in China, where roughly 71.95 percent of grandparents help to raise grandchildren, increasing to an even higher share in large cities ([51]).

Scholarly discourse presents a dual narrative regarding co-parenting outcomes ([25]). On the positive side, grandparental care can reduce maternal stress, enhance child adjustment, and give grandparents a renewed sense of purpose ([25]; [29]). Conversely, intensive involvement has also been linked to greater anxiety and stress for both generations, with downstream risks for children’s well-being ([5]; [7]). Most existing evidence, however, is drawn from Western samples. China presents a markedly different context: Confucian filial-piety norms place grandparents at the apex of the family hierarchy, and adult children are expected to defer to their elders’ child-rearing views ([25]). Such obligatory ties differ markedly from the “non-interference” ethos common in bilineal Western families ([11]).

As grandparental participation grows, families move from a two-generation to a three-generation system that must be reorganized ([27]). Family systems theory suggests that grandparental involvement increases triangulated relationships within the family, amplifying parental stress ([21]). Parenting stress is a form of psychological distress that arises when caregiving requirements exceed parents’ coping resources, undermine parental well-being, disrupt parenting practices, and impede children’s development ([27]).

Early childhood is a particularly sensitive period. About 13 percent of U.S. mothers with three-to-five-year-olds children experience chronically high parenting stress ([17]); the figure climbs to 28 percent among Chinese parents of preschoolers ([15]). Although overall stress levels appear to be comparable in China and Japan, the specific stressors and coping patterns differ ([31]), underscoring the need for analyses firmly grounded in China’s unique cultural context. In Chinese families where multiple generations share caregiving responsibilities, the intertwining of hierarchical norms and divergent child-rearing beliefs may amplify both parenting stress and its antecedents.

Despite the size and cultural distinctiveness of China’s parent–grandparent co-parenting families, few studies have examined how parenting stress is patterned within this group or what factors predict heightened stress. Addressing this gap is critical for designing culturally responsive interventions that support parental well-being and, by extension, healthy child development. Accordingly, the present study investigates (1) latent profiles of parenting stress in Chinese parent–grandparent co-parenting families and (2) parental, social–contextual, and child-level predictors of profile membership, thereby extending co-parenting research beyond Western contexts and illuminating the unique dynamics of multigenerational caregiving in contemporary China.

## 2. Literature Review

### 2.1. Theoretical Framework

Family systems theory, originally proposed by Bowen, views the family as an interconnected emotional unit, where stress experienced by one member often transmits throughout the system. Specifically, triangulation describes how parental stress is transferred within family relationships, with anxiety shifting among family members ([21]). For example, when parents experience stress, it can be absorbed by other family members, temporarily relieving the parent but potentially passing tension onto the child. This cyclical process may lead to the child developing symptoms that, in turn, escalate parental anxiety, creating a vicious cycle. In larger family systems, such as parent–grandparent co-parenting families, the number of potential triangulation interactions increases, amplifying the complexity of emotional exchanges. In the context of parent–grandparent co-parenting, the entry of grandparents into the caregiving dynamic creates additional triadic interactions. While these new relational ties can offer support, they may also exacerbate stress if conflicts arise over parenting approaches or caregiving responsibilities. As anxiety circulates within the family system, it can adversely affect parental well-being and, ultimately, child development. This underscores the importance of examining how the family system, when expanded through grandparental involvement, reshapes the experience of parental stress. Complementing FST, the parenting process model (PPM) provides a useful lens for understanding how parental stress develops within complex family environments (see Figure 1). The PPM emphasizes that parenting is shaped by the interplay of factors across three domains: parental characteristics, social–contextual factors, and child factors ([9]). Applying this framework to parent–grandparent co-parenting arrangements offers an opportunity to examine how individual and contextual factors intersect to shape parental stress in these increasingly common family structures.

### 2.2. Influential Factors of Parental Stress

Building upon the theoretical foundations outlined above, this section delves into specific factors that may shape parenting stress within parent–grandparent co-parenting families. Guided by the parenting process model and prior empirical findings, we focus on parental characteristics, social–contextual factors, and child-level predictors particularly relevant in the Chinese sociocultural context.

#### 2.2.1. Parenting Self-Efficacy

Parenting self-efficacy is a parent’s perception of parenting’s ability to positively influence the behavior and development of their children ([6]). In recent years, an increasing number of scholars have focused on the impact of parenting self-efficacy on parenting stress ([47]), such as a year-long longitudinal study of 65 mothers of children that indicated that higher self-efficacy in mothers of young children was associated with lower parenting stress ([47]). Researchers also found that mothers with high parenting self-efficacy were less likely to be in a group with chronically high and increased parenting stress ([14]). However, the study focused only on the parenting stress of mothers of young children; fathers, as important players in the family, should also be considered. Based on this, through a survey of Chinese parents with one and two children, Hong et al. found that parents with higher parenting self-efficacy were more likely to experience lower parenting stress ([27]). In addition, it has been suggested that parenting self-efficacy positively predicts supportive parenting behaviors ([13]) and that positive parenting behaviors lead to less parenting stress ([32]). While these studies are important for exploring parenting stress, they have not focused on the effects of parenting self-efficacy on different types of parenting stress among parent–grandparent co-parenting families. Therefore, this study proposes the following research hypothesis:

**Hypothesis** **1.**
*Families with higher parenting self-efficacy are more likely to be categorized in the lower parenting stress profile.*


#### 2.2.2. Co-Parenting Relationship

The concept of parent–grandparent co-parenting was expanded from that of parental co-parenting ([35]). In this study, parent–grandparent co-parenting was defined as, in new or primary families, grandparents assuming some parenting responsibilities and co-parenting with parents to raise the third generation ([35]). Increasingly, researchers are focusing on the effects of the parent–grandparent co-parenting relationship on parenting stress in young children. Li et al. used Amos structural equation modeling to find that a harmonious parent–grandparent co-parenting relationship reduced scores on the dimensions of parenting distress and parent–child dysfunctional interaction for maternal parenting stress ([35]). In addition, social support includes emotional support and social network support (e.g., support from relatives or neighbors), and social support has a major impact on parenting stress in mothers of young children, with low social support and negative life events leading to more parenting stress ([45]). As an important part of social support, grandparent support is likely to influence parenting stress. Unfortunately, although some scholars have explored the effects of grandparents’ parenting support (co-parenting support is one of the dimensions of the Parent–Grandparent Co-parenting Relationship Scale) on parenting stress ([33]), these studies have only looked at parenting stress as a homogeneous whole to explore the effects of the parent–grandparent co-parenting relationship on parenting stress. Therefore, this study proposes the following research hypothesis:

**Hypothesis** **2.**
*Families with a better co-parenting relationship are more likely to be categorized in the lower parenting stress profile.*


#### 2.2.3. Co-Parenting Structure

Beyond relational quality, the specific structure of the co-parenting structure, particularly whether caregiving is provided by maternal or paternal grandparents, may further influence parenting stress. Research has shown that different co-parenting structures can affect parenting stress ([19]). For example, [19] ([19]) found through interviews that paternal grandparents may be a source of stress for new mothers whose in-laws blame them for being unable to produce a male child, and this idea of preferring boys over girls can create invisible pressure for new mothers. In addition, using interviews and survey data, [59] ([59]) found that while paternal grandmothers living with mothers were positively associated with maternal stress, support from maternal grandmothers tended to reduce maternal parenting stress ([59]). For this reason, mothers were in a stronger bargaining position when dealing with maternal grandmothers than with paternal grandmothers. The benefit of providing care that meets the mother’s requirements and allows the mother to express her requests comfortably is that it can reduce parenting stress. In summary, these studies have found that paternal grandparent co-parenting contributes to greater parenting stress among mothers of preschool children. However, they have ignored the influence of paternal grandparents on parenting stress among fathers of preschool children and the effects of the co-parenting structures on parenting stress in different characteristic groups. Therefore, this study proposes the following research hypothesis:

**Hypothesis** **3.**
*Families in which the maternal grandparents co-parent are more likely to be categorized in lower parenting stress profiles.*


#### 2.2.4. Family Income

Economic resources represent another crucial contextual factor. Based on the Family Stress Model, economic hardship can negatively impact parental mental health, and parents with low family income may experience higher stress levels ([38]). The study of Gilsook and Jahng confirms this view, as they conclude that the greater the financial burden on the mother, the more likely it is to lead to increased parenting stress ([22]). In addition, [46] ([46]) suggest that higher levels of parenting stress may occur when families are financially stressed ([46]). These studies suggest that family income affects mothers’ parenting stress, but these studies do not focus on the effect of family income on parenting stress in parent–grandparent co-parenting families. Based on this, this study proposes the following research hypothesis:

**Hypothesis** **4.**
*Families with lower incomes are more likely to be categorized with higher parenting stress profiles.*


#### 2.2.5. Gender of Children

Parenting stress is influenced not only by family factors but also by individual factors in young children ([1]). In family systems theory, young children have gained extensive attention from researchers as the ultimate recipients of anxiety. The gender and age of young children may influence parenting stress ([52]). In terms of gender, previous research surveying 90 parents of 3-month-old infants found that mothers of sons reported more stress than mothers of daughters. Some researchers have explored the causes of such phenomena and found no difference in rates of disruptive behavior problems between girls and boys at age 2, but more girls showed a decrease or no increase in disruptive behavior in early childhood ([28]). In addition, Williford’s assessment of 430 young children at the ages of 2, 4, and 5 years found that higher levels of externalizing behavior problems predicted higher parenting stress in toddlerhood ([52]). Moreover, one study found that boys’ higher levels of hyperactivity may cause more parental stress ([8]). These studies show that gender is an important factor influencing parenting stress. Therefore, this study proposes the following research hypothesis:

**Hypothesis** **5.**
*Families raising boys are more likely to fall into the higher parenting stress profile.*


#### 2.2.6. Age of Children

Finally, it is also important to examine the role of gender in young children as a predictor of parenting stress ([52]). In recent years, a growing number of researchers have focused on the impact of preschool children’s age on parenting stress. For example, previous studies have found that parenting stress among mothers of young children may decrease with the age of the child ([42]; [52]). Previous studies have explained why this phenomenon occurs. One study found that many children’s externalizing problems decline with age in preschool years ([12]), and a decrease in a child’s externalizing problems may reduce parenting stress. However, previous studies have paid less attention to the predictive role of young children’s age in parenting stress in parent–grandparent co-parenting families. Based on this, this study proposes the following research hypothesis:

**Hypothesis** **6.**
*Families with older children are more likely to fall into the lower parenting stress profile.*


### 2.3. Strengths of a Latent Profile Analysis Approach

Although a few previous studies focused on parenting stress among parents of young children, most of these studies examined parenting stress as a whole and used a variable-centered perspective to explore levels of parenting stress ([27]). However, parenting stress may be heterogeneous due to inter-individual variability. As a person-centered classification method, latent profile analysis (LPA) adopts probability-based mixture modeling. It assumes that individuals can be classified according to a pattern of comparable characteristics ([10]). LPA can better help us identify various traits in heterogeneous samples ([43]). In large-sample studies, the fit metrics of latent profile analysis can ensure the accuracy of the classification, and its classification is better than that of traditional cluster analysis ([54]). Based on this, researchers conducted a potential profile analysis of parenting stress among parents with one and two children. They found that parenting stress among young children was classified as low, moderate, or high stress ([27]). In addition, researchers tracked the parenting stress trajectories of 580 mothers of young children from 14 months to 36 months of age. They found that mothers of young children were a heterogeneous group and divided parenting stress into three trajectory categories: a chronically high group (7% of the sample), an increasing group (10% of the sample), and a decreasing group (83% of the sample) ([14]). However, none of these studies have focused on whether there is heterogeneity in parenting stress in parent–grandparent co-parenting families.

With the aging of China’s population and the advent of the three-child era, the number of parent–grandparent co-parenting families will increase, and for this reason, it is necessary to consider the variables as interdependent systems from a person-centered perspective based on family systems theory, classify the parenting stress of parents of young children into different types according to the multiple profiles of parenting stress in parent–grandparent co-parenting families, and explore the effects of predictors on different profiles of parenting stress.

### 2.4. The Present Study

After analysing the existing relevant studies, the current research attained the following research scope: First, most prior studies have been conducted in Western contexts, overlooking the unique culture of parent–grandparent co-parenting in China, where Confucian traditions and hierarchical family structures may intensify parenting stress. Second, although existing research has explored the general relationship between grandparental involvement and parenting stress, little attention has been given to the heterogeneity of parenting stress profiles within multigenerational families. Specifically, the field lacks person-centered analyses that capture the diverse patterns of stress experienced by parents in grandparent-involved households. Third, while individual predictors of parenting stress, such as parental efficacy, co-parenting relationships, family income, and a child’s sex and age, have been examined in isolation, few studies have integrated these factors within a comprehensive framework to explore how to predict distinct profiles of parenting stress in Chinese parent–grandparent co-parenting families. Addressing these gaps will not only enrich our theoretical understanding of family functioning in complex caregiving systems but also inform culturally sensitive interventions to support parental well-being and child development.

In the current study based on family systems theory, a parenting process model was used to test (1) whether there are distinct profiles of parenting stress among parent–grandparent co-parenting families in China based on the Parenting Stress Index-Short Form Questionnaire, a well-established measure of parenting stress in parents of young children ([50]), and (2) whether different parental characteristics (parenting self-efficacy), social–contextual factors (parent–grandparent co-parenting relationship, co-parenting structure, family income), and child factors (gender, age) can predict these profiles, which will be critical for informing the development of targeted support and intervention strategies tailored to varying parenting stress patterns.

## 3. Materials and Methods

### 3.1. Participants and Procedure

A total of 317 parents involved in parent–grandparent co-parenting participated in this study with an anonymous questionnaire, and 14 parents were excluded from the sample due to errors in completing the questionnaires. The final sample included 303 parents of preschool children (Table 1). A total of 159 (52.5%) parents of sons and 144 (47.5%) parents of daughters were enrolled in this study. The mean age of these children was 61.17 months. Of the participating families, 245 (80.9%) of the questionnaire respondents were mothers of young children, and 58 (19.1%) were fathers of young children. A total of 227 (74.9%) of grandparents living with parents and raising young children together were paternal grandparents, and 76 (25.1%) were maternal grandparents. A total of 63 (20.8%) of the co-parenting grandparents were male and 240 (79.2%) were female. Among the parents of young children, 282 (93.1%) of the mothers of young children were primary caregivers, and 21 (6.9%) of the fathers of young children were primary caregivers. The per capita monthly income of the parents of young children in most of the families in this study was RMB 5001–10,000 (see Table 1).

To explore parenting stress and its influencing factors among Chinese parents in joint parent–grandparent families, and taking into account the geographical differences between southern and northern China, this study used a convenience sampling method. Participants came from three cities in southern China and three cities in northern China, and they were parents of young children in parent–grandparent co-parenting families, and their children were in kindergarten. Parents were recruited to participate in this study in six kindergartens, and they were briefed by the kindergarten teacher on the study procedures, the fact that the data were anonymous, and the purpose of this study, as well as informed that the researcher would provide one-on-one parenting feedback reports for parents who left their e-mail address. Parents who volunteered to participate in this survey used the online questionnaire platform to answer the questions. Participants took about 20 min to complete the questions. The study protocol was reviewed and approved by the ethics committee of Central China Normal University, and all parents who participated in this study gave their consent for their data to be used in the research study.

### 3.2. Measurement

#### 3.2.1. The Parenting Stress Index-Short Form (PSI-SF)

The PSI-SF Third Edition (PSI-SF; [3]) is a 36-item self-report questionnaire used to investigate parenting stress among parents of children aged 1 month to 12 years old. The questionnaire includes three subscales (parental distress [PD], parent–child dysfunctional interaction [P-CDI], and difficult child [DC]) ([4]), and each of the three subscales consists of 12 items. The scale is based on a five-point Likert scale ranging from “strongly disagree” to “strongly agree”, with scores ranging from 1 to 5, with higher scores indicating higher parenting stress. Parenting stress levels were categorized according to the total score of the Parenting Stress Questionnaire: a score ≤ 85 indicating a normal standard of parenting stress; a score of 86–90 indicating some parenting stress but not too much; a score of 91–98 indicating a high level of parenting stress; and a score ≥ 99 indicating a very high level of parenting stress ([2]). The scale is highly reliable and valid in the Chinese cultural context ([57]). In the present study, Cronbach’s α was 0.882, 0.887, and 0.893 in that order.

#### 3.2.2. The Co-Parenting Relationship Scale (CRS)

The CRS developed by [18] ([18]) and revised by [34] ([34]) was used to measure the parent–grandparent co-parenting relationship and the revised co-parenting relationship scale was reliable and valid in the Chinese cultural context. This 38-item scale uses a seven-point scale ranging from “very non-compliant” to “very compliant” on a scale of 1 to 7. The CRS consists of 38 items in seven dimensions of parenting consistency (4 items), parenting closeness (5 items), conflict exposure (5 items), parenting support (6 items), parenting approval (7 items), parenting sabotage (6 items), and division of labor (5 items). The two dimensions of conflict exposure and parenting sabotage were reverse-scored, with higher scores on each dimension representing a more positive self-report on that dimension and higher total scores on the questionnaire representing a better parent–grandparent co-parenting relationship. In this study, Cronbach’s α was 0.733, 0.778, 0.873, 0.890, 0.836, 0.800, and 0.801 in that order.

#### 3.2.3. Parenting Self-Efficacy (PS-E)

This study used the Parenting Self-Efficacy Scale developed by [36] ([36]) to measure parenting self-efficacy, which includes three dimensions of caregiving efficacy (6 items), emotional communication efficacy (7 items), and parenting confidence (6 items). The scale has 19 questions and is scored on a five-point Likert scale from 1 to 5, ranging from “very non-compliant” to “very compliant”, with higher scores indicating higher parenting efficacy. In this study, Cronbach’s α of the scale was 0.890, 0.866, and 0.911 in that order.

### 3.3. Data Analysis

Descriptive statistics and multinomial logistic regression were performed in SPSS 26.0. Latent profile analysis (LPA) was conducted using Mplus 8.3 to assess different characteristics of parenting stress, explore the most likely number of characteristics of parenting stress (optimal model), evaluate five LPA models (from I to V characteristic models), and compare the fit criteria of the different models to help determine which latent profile model was the most accurate ([43]). After determining the optimal number of latent profiles, multinomial logistic regression was conducted using SPSS 26.0 to assess how parental characteristics (parenting self-efficacy), social–contextual factors (parent–grandparent co-parenting relationship, co-parenting structure, family income), and child factors (gender, age) predict different profiles of Chinese parents based on parenting stress.

## 4. Research Results

### 4.1. Latent Profile Analysis

AIC, BIC, ABIC, and Entropy were taken into consideration in the current study to select the most appropriate profile. Table 2 shows these values for each profile. Researchers usually start with models associated with hypotheses, a priori predictions, substantive findings, and fit indices. Smaller values of AIC, BIC, and ABIC indicate better model fit results ([39]). A larger Entropy indicates higher correctness of the model classification. In this study, the AIC and ABIC values of the models with type IV and V profiles were lower than those of the other models, but LMR-LRT and BLRT were not significant in the models with type V profiles, and the Entropy value was the largest for type IV. In addition, in solutions with type IV profiles, each profile contained >5% of the sample ([41]). Given these considerations, the solution with type IV profiles was considered the best model.

To test the accuracy of the classification results of potential profiles, discriminant analysis was conducted using the three dimensions of parenting stress as indicators. Analysis of variance found that the very low parenting stress group scored significantly lower than the low parenting stress group, high parenting stress group, and very high parenting stress group on the three dimensions. The high parenting stress group scored significantly lower than the very high parenting stress group on all three dimensions, indicating heterogeneity in the potential categories of different types of parenting stress (see Table 3). Therefore, the four-profile solution was chosen as the optimal model in this study.

The four-profile solution is shown in Figure 2. As can be seen, participants in Profile 1 (12.9%, n = 39) had the lowest value for each item of the PSI-SF scores (M = 1.492, SD = 0.391) and were labeled the “low parenting stress group”. Participants in Profile 2 (39.3%, n = 119) were characterized by having middle scores of each item of PSI-SF and were labeled the “middle parenting stress group” (M = 2.108, SD = 0.362). Participants in Profile 3 (40.6%, n = 123) were characterized by having relatively high scores on each item of the PSI-SF subscale and were labeled the “high parenting stress group” (M = 2.714, SD = 0.397). Participants in Profile 4 (7.2%, n = 22) were characterized by having the highest scores on each item of the PSI-SF subscale, and the total score of each questionnaire was greater than or equal to 99, indicating a very high level of parenting stress ([2]). Therefore, these participants were labeled the “very high parenting stress group” (M = 3.346, SD = 0.475).

### 4.2. Predictors of Latent Profile Membership

Based on latent profile analysis, the factors influencing the potential categories of parenting stress were further explored. We conducted multinomial logistic regression to explore the effects of parental characteristics (parenting self-efficacy), social–contextual factors (parent–grandparent co-parenting relationship, co-parenting structure, family income), and child factors (gender, age) on the profile classification of parenting stress. Table 4 summarizes the results of these analyses.

The high parenting stress group had lower scores for co-parenting relationships, parenting self-efficacy, and family income than the middle parenting stress group. When maternal grandparents were used as a reference, paternal grandparents were more likely to be in the very high parenting stress group than in the low parenting stress group. Specifically, when using the low parenting stress group as a reference group, co-parenting relationships negatively predicted the other three groups, with poorer parenting relationships more likely to be attributed to the high parenting stress group, the next most likely to be attributed to the very high parenting stress group, and the least likely to be attributed to the middle parenting stress group. The co-parenting relationship positively predicted the low parenting stress group and middle parenting stress group when the very high parenting stress group was used as a reference group. In other words, the better the co-parenting relationship, the more likely it was to be categorized in the low parenting stress group and the middle parenting stress group rather than the very high parenting stress group. When the low parenting stress group was used as the reference group, parenting self-efficacy negatively predicted the other three groups. In other words, the higher the parenting self-efficacy, the more likely it was to be classified in the low parenting stress group rather than the other three groups. When the low parenting stress group was used as the reference group, the lower the family income, the more likely it was to be classified in the high parenting stress group and the very high parenting stress group. When the very high parenting stress group was used as the reference group, the higher the income, the more likely it was to be classified in the low parenting stress group, with the next highest probability of being ranked in the middle parenting stress group.

Male preschoolers were more likely to be in the high parenting stress group than in the low parenting stress group compared to female preschoolers. However, there was no significant effect of preschoolers’ age on the four profiles of parenting stress.

## 5. Discussion

Although previous studies have shown heterogeneity in parenting stress ([27]), few empirical studies have explored patterns of parenting stress and its associated factors, particularly in parent–grandparent co-parenting families in China. The current study aims to examine the latent profiles of parenting stress among parent–grandparent co-parenting families in China to fill the gap created by the lack of previous research. These characteristics were determined using LPA, and they are considered to be one of the best methods to determine individual heterogeneity ([55]). As a result, four profiles with distinct characteristics were identified: a low parenting stress group, a middle parenting stress group, a high parenting stress group, and a very high parenting stress group. Furthermore, the distinct profiles were associated with differences in parenting self-efficacy, co-parenting structure, family income, parent–grandparent co-parenting relationship, and the gender of young children. The results of the study are discussed in more detail below.

### 5.1. Profiles of Parenting Stress

The LPA was used in the current study to explore parenting stress among parent–grandparent co-parenting families in China, and we found considerable heterogeneity in parenting stress. An optimal four-profile solution was generated in this sample. In particular, the results of the data from all four profiles showed that parental distress had the highest scores and that parent–child dysfunctional interaction had the lowest scores in PSI-SF. This suggests that parenting stress in parent–grandparent co-parenting families is more likely due to parental distress than to parent–child dysfunctional interaction. Specifically, in Profile 1, the low parenting stress group, the total PSI-SF questionnaire score was 53.712, which is in line with [2]’s ([2]) categorization of the normal standard of parenting stress (with a score ≤ 85). Profile 1 comprised 12.9% of the parents in this study, which was the second lowest percentage of the four profiles. The results of this profile illustrate that fewer parent–grandparent co-parenting families fall into the low parenting stress group.

Profile 2 was the middle parenting stress group. In this profile, the total PSI-SF questionnaire score was 75.888, which was relatively lower than [2]’s ([2]) definition of the normal parenting stress criteria (with a score ≤ 85). Profile 2 included 39.3% of the sample, which was the second-highest percentage of the four profiles. The results of this profile suggest that parents in more parent–grandparent co-parenting families have middle levels of parenting stress.

Profile 3 was the high parenting stress group. In this profile, the total PSI-SF questionnaire score was 97.704, which was consistent with a high level of parenting stress (with a score of 91–98), as defined by [2] ([2]). Profile 3 included 40.6% of the sample, which was the highest percentage of the four profiles. The results of this profile suggest that most parent–grandparent co-parenting families are characterized by a high level of parenting stress.

Profile 4 was the very high parenting stress group. In this profile, the total PSI-SF questionnaire score was 120.456, which was consistent with a very high level of parenting stress (with a score ≥ 99), as defined by [2] ([2]). Profile 4 included 7.2% of the participants, which was the lowest percentage of the four profiles. This result is consistent with that of [54] ([54]), who reported that individuals scoring high on all parenting stress items had the smallest sample size ([54]). This group reported the highest overall level of parenting stress of the four groups. Moreover, subscale 3 (difficult child) was significantly higher than the rest of the profiles. Thus, difficult children may be a key indicator of high parenting stress among Chinese parents. In contrast, scores on subscale 2 (parent–child dysfunctional interaction) did not stand out as significantly higher than the scores on the other profiles, suggesting that Profile 4 is more likely to be affected by difficult children than by parent–child dysfunctional interactions leading to greater parenting stress. This finding is consistent with a previous longitudinal study that showed that higher levels of externalizing behavior problems, anger tendencies, and emotional dysregulation were associated with greater parenting stress in early childhood ([52]).

The results of the four-profile solution extend an earlier study on Chinese one- and two-child families, which yielded only three stress classes ([27]). A likely explanation is that multigenerational households embed parents in a denser web of relationships. Family systems theory suggests that the involvement of grandparents increases the number and complexity of triangulated interactions within the family system. This expansion multiplies opportunities for role negotiation and authority clashes in Chinese families. Consistent with this reasoning, the mean PSI-SF item score in our parent–grandparent sample (M = 2.37) exceeded the mean score reported for non-co-parenting (M = 2.16) families ([53]), and the “high parenting stress” profile encompassed 40.6 % of respondents. Sociocultural factors further illuminate why parent–grandparent co-parenting families were over-represented in the “high parenting stress”. Under patrilineal Confucian norms, paternal grandparents have greater hierarchical authority and stronger expectations regarding lineage continuity and child-rearing practices. When these expectations collide with the autonomy-oriented views of urban, dual-earner parents, parental stress is likely to increase. In contrast, maternal-grandparent involvement often grants mothers more bargaining power and value alignment, reducing stress—a pattern repeated in qualitative studies showing that mothers feel more “heard” when supported by their own parents ([26]).

In short, the additional triadic ties described by family systems theory, coupled with China’s hierarchical kin norms, help to explain why parent–grandparent co-parenting families yield a more complex—and often more stressful—parenting landscape than that observed in prior nuclear-family studies.

### 5.2. Factors Influencing Parenting Stress

The results showed that the age of young children did not significantly affect the four profiles of parenting stress, and there were significant effects of the parent–grandparent co-parenting relationship, parenting self-efficacy, parenting structure, family income, and gender of preschool children on the profiles of parenting stress. The results of this study confirm and extend, to some extent, the hypothesis of the parenting process model, namely that parenting stress is directly influenced by forces emanating from three subsystems (i.e., parent, social context, child) in parent–grandparent co-parenting families. However, parenting stress is not affected by the age of young children. This study also supports the idea in family systems theory that family members are interdependent and that family subsystems are influenced by other subsystems in the home. The specific analysis of the results is provided below.

Participants with a more positive parent–grandparent co-parenting relationship were significantly more likely to be in the low parenting stress group than in the other three profiles of parenting stress. This finding is supported by [35] ([35]), who found that a positive parent–grandparent co-parenting relationship helps to reduce the parenting stress of mothers. In a positive parent–grandparent co-parenting relationship, grandparents provide positive social support for parents, which can help parents reduce the stress of parenting ([33]). More social support for parents means that parents receive more emotional and practical help from others, and this outside help can help to improve parents’ ability to cope with parenting challenges and ease parenting stress ([37]). Therefore, reducing parent–grandparent co-parenting conflicts and disagreements and creating harmonious parenting relationships can reduce parenting stress. According to family systems theory, when families are in a tense relationship, it is important to try to remain stable and in the “onlooker” position. People who are on the sidelines are more emotionally stable and able to think rationally. This approach can also benefit the entire family. Specifically, once a family member knows that they are in a tense relationship, it is important to first engage the other two family members in an emotionally neutral manner; secondly, it is important to trust that the other two family members are capable of solving the problem and thinking calmly; and finally, it is important to ask more factually relevant questions ([21]). Thus, family members can mitigate the negative effects of a strained parent–grandparent co-parenting relationship on the family in this way.

This study found that the higher the parenting self-efficacy, the more likely it was to be classified in the low group compared to the other three groups of parenting stress. This finding is supported by a study comparing parenting stress among parents of one and two children in China, which found that parenting self-efficacy reflects parents’ perceptions of their ability to perform competently in their parenting role and that parents with high parenting efficacy were more confident in their parenting abilities, showed more positive emotions when faced with parenting challenges, and experienced fewer feelings of parenting stress than parents with low parenting efficacy ([27]). In addition, another study has found that self-efficacy reduces the relationship between income and parenting stress ([47]), meaning that high-income parenting plays an extremely important role in relieving parenting stress. Therefore, drawing on Bandura’s theory of self-efficacy, parents of young children can take the initiative to acquire parenting knowledge from multiple sources and intentionally train and improve their parenting skills. On the other hand, parents’ relatives and friends, kindergartens, communities, or specialized family education guidance services should also show trust and positive and encouraging attitudes, explore the strengths of parents of young children, and provide them with relevant experience or professional guidance, as well as targeted psychological guidance, on a regular basis to empower parents.

Paternal grandparents were more likely to be classified in the very high parenting stress group than in the low parenting stress group compared to maternal grandparents. This finding is confirmed by previous studies. A study of 372 preschool children and their families in China found that maternal grandparents in co-parenting families scored significantly lower on the dimension of conflict exposure (conflict exposure is one of the dimensions of the Parent–Grandparent Co-parenting Relationship Scale) than paternal grandparents ([35]). Previous evidence further indicates that strained co-parenting relationships amplify parenting stress ([23]). Why might paternal grandparent care be uniquely stressful? Two interlocking sociocultural mechanisms are noticeable here. On one hand, the primary caregiver role of mothers may play a role. In our sample, 93.1% of mothers were the main day-to-day caregivers. When caregiving authority must be negotiated with paternal grandparents, especially with mothers-in-law, mothers face both generational and hierarchical gaps in child-rearing beliefs. Moreover, the conflicts between mothers-in-law (grandmothers) and daughters-in-law (mothers) are constant in the traditional Chinese cultural context ([48]). Therefore, these conflicts and contradictions may lead to parents feeling greater parenting stress in families with parent–grandparent co-parenting. Such clashes are far less frequent with maternal grandparents, with whom mothers share stronger emotional bonds and greater value alignment ([58]). On the other hand, communication quality can play a role. [26] ([26]) showed that open, respectful communication predicts cooperative co-parenting, whereas low-quality dialog allows minor disagreements to spiral into chronic conflict. In patrilineal households, fathers often act as cultural “bridges”, yet dual-earner fathers may have limited time to mediate. Without this coordination, unresolved disagreements accumulate and push families into the very high-stress bracket. To avoid negative effects caused by the structure of parenting, first, interventions should focus on equipping mothers and paternal grandparents with effective conflict-resolution skills to better manage intergenerational disagreements. Second, it is essential to encourage fathers to actively assume the role of “facilitator” within the family, helping to mediate between generations and promote harmonious caregiving dynamics. Finally, community-based workshops that promote egalitarian decision-making and open communication can further transform potentially contentious parent–grandparent co-parenting families into more supportive partnerships.

When the very high parenting stress group was used as a reference group, the higher the income, the more likely it was to be classified in the low parenting stress group, and the next most likely it was to be classified in the middle parenting stress group. The Family Stress Model supports this conclusion, suggesting that financial hardship can negatively impact parental mental health and that parents with low family incomes may experience higher stress levels ([38]). In addition, economic stress creates an adverse developmental environment that not only increases the likelihood of problem behaviors in young children but also has a significant impact on children’s cognitive and language development ([40]), and parents may experience higher levels of parenting stress due to problem behaviors and slow growth. In view of this, first, China can learn from the best experiences of other countries and, taking into account the country’s current situation, provide varied services and support for low-income families. Second, the Perry Preschool Program Study shows that preschool education is one of the most cost-effective public investments with the highest rate of return. Therefore, it is absolutely necessary to adhere to the principle of public welfare and increase funding for preschool education. For example, the state can provide more financial support for preschool education by setting up a separate education lottery. Finally, we should strengthen the standardization, transparency, and fairness of managing preschool education funds to ensure that education funds are really taken from the people and used for the people to provide varied support for low-income families.

Parents of sons were more likely to be in the high parenting stress group than in the low parenting stress group compared to daughters. This result aligns with findings from Western contexts, where boys are more likely to exhibit externalizing behaviors, such as aggression, impulsivity, and hyperactivity, which may elevate parenting stress during preschool years ([52]; [8]). However, in China, these behavioral tendencies may be further exacerbated by sociocultural dynamics, notably traditional son preference. Specifically, grandparental involvement tends to be gender-biased, with grandparents prioritizing investment in grandsons over granddaughters ([16]). While this support may be intended to benefit the family, it can inadvertently create role conflict: when grandparents overindulge sons or contradict parental disciplinary strategies, parents may feel undermined, leading to increased daily friction and heightened stress. Thus, the combination of boys’ behavioral traits and culturally reinforced gendered expectations contributes to a heavier parenting burden in Chinese parent–grandparent co-parenting families raising sons. Therefore, it is also important for grandparents to abandon the traditional patriarchal concept of not being overly involved in parenting and to clarify their role as facilitators, as parents are the primary providers of early childhood education.

Notably, this study found that the age of young children did not significantly predict membership of any of the four parenting stress profiles, a finding that diverges from earlier research. For example, [52] ([52]) reported that as children mature, their growing behavioral independence tends to alleviate parental stress. However, these prior studies primarily focused on nuclear families and did not consider multigenerational caregiving dynamics. In parent–grandparent co-parenting families, grandparents often assume substantial caregiving roles, protecting parents from the daily burdens associated with caring for younger, more dependent children ([24]). This effect may help to explain why the child’s age did not emerge as a significant stressor in our sample. More broadly, this result underscores the dual nature of grandparental involvement. On the one hand, grandparents can provide practical and emotional support, relieving time pressure and easing parental stress. On the other hand, as earlier sections of this study have highlighted, intensive grandparental involvement, especially when coupled with hierarchical authority structures and intergenerational conflicts, can itself become a source of parental tension. This complex interplay suggests that the parenting process model, which emphasizes child-driven stressors, may have limited applicability in the unique context of Chinese parent–grandparent co-parenting families.

### 5.3. Limitations and Future Directions

The limitations of the present study should be noted. First, this study relied solely on parental report data, overlooking the direct perspectives of grandparents despite their central role in co-parenting. Without including grandparental viewpoints, such as their beliefs, caregiving styles, and attitudes, our understanding of parent–grandparent co-parenting dynamics remains incomplete. Future research should explicitly collect data from both parents and grandparents to reduce bias and discover how intergenerational expectations influence parenting stress. Second, this study did not determine which factor had the greatest influence on parenting stress in parent–grandparent co-parenting families. According to the parenting process model, parent personality may have a greater impact, but there is still a lack of data support from empirical studies. Future research can compare the degree of influence of the three subsystems on parenting stress in parent–grandparent co-parenting families based on the parenting process model. Moreover, other aspects of the three subsystems were not explored, such as the roles of marital relationships and personality characteristics in the parenting subsystem, friend support and occupational experience in the social subsystem, and temperament type and externalized behavioral problems in the early childhood subsystem, all of which may affect parenting stress in parent–grandparent co-parenting families. Additionally, the family system needs to be considered in a broader social context. Therefore, more cultural and historical factors need to be considered in future research. Third, some researchers have noted that grandparents provide a positive form of social support for parents that can reduce maternal parenting stress ([33]). Therefore, future research could compare differences in parenting stress between families with co-parenting and those with parent–grandparent co-parenting.

## 6. Conclusions and Implications

To the best of our knowledge, this study is the first to apply latent profile analysis (LPA) to a sample of Chinese parents in parent–grandparent co-parenting households, identifying distinct profiles of parenting stress based on specific PSI-SF items rather than total scores. We identified four parenting stress profiles: low, moderate, high, and very high-stress groups. Furthermore, parenting self-efficacy, co-parenting structure, family income, parent–grandparent co-parenting relationship, and child gender were significant predictors of profile membership, revealing important variations in parental stress within these multigenerational families. These findings underscore the heterogeneity of parenting stress in Chinese parent–grandparent co-parenting families and support the applicability of family systems theory and the parenting process model in this context. However, the non-significant role of child age suggests that the parenting process model may require refinement when applied to multigenerational caregiving structures, where shared responsibilities potentially reduce age-related variations in child-rearing demands.

Importantly, these findings also offer concrete implications for practice and policy. First, considering the higher likelihood of parent–grandparent co-parenting families falling into the high parenting stress profile (40.6%), targeted family counseling interventions could help to mitigate conflicts rooted in traditional hierarchical structures. For example, evidence from the Family Relationship Centre at Broadmeadows in Australia highlights how culturally sensitive family counseling services can effectively mitigate intergenerational conflicts within multicultural families, helping to navigate complex caregiving roles and reduce parental stress ([44]). Implementing similar evidence-based family support services in China could empower parents, particularly mothers, to negotiate caregiving roles more effectively. Second, our findings regarding child gender suggest that families raising boys may benefit from parenting workshops addressing behavioral regulation and emotional coaching. Research from cross-cultural contexts, including German ([49]), European American, African American, biracial, and Hispanic families, has shown that early parenting interventions focused on managing boys’ externalizing behaviors can reduce parental stress and improve child outcomes ([52]). Third, the study’s findings call for theoretical advancement. While family systems theory and the parenting process model provided valuable perspectives for understanding the multilayered stress dynamics in parent–grandparent co-parenting families, our results also expose their limitations in this cultural context. Specifically, the non-significant effect of child age suggests that multigenerational caregiving structures may buffer certain stressors traditionally emphasized in nuclear family models. Future theoretical refinements should integrate cultural variables, such as Confucian hierarchical norms, son preference, and grandparental caregiving roles, to better capture the complexities of Chinese family systems. Moreover, empirical research should further explore the interplay between family subsystems, extending beyond individual traits to include marital quality, occupational stress, and grandparents’ caregiving philosophies. Such culturally grounded extensions would not only enrich theory but also enhance the precision of interventions aimed at reducing parenting stress in diverse family constellations.

In sum, this study advances both theory and practice by illuminating the stress profiles and influencing factors among Chinese parent–grandparent co-parenting families. It provides a critical foundation for developing culturally sensitive frameworks and interventions that address the unique challenges faced by multigenerational households in contemporary China.

## Figures and Tables

**Figure 1 behavsci-15-00533-f001:**
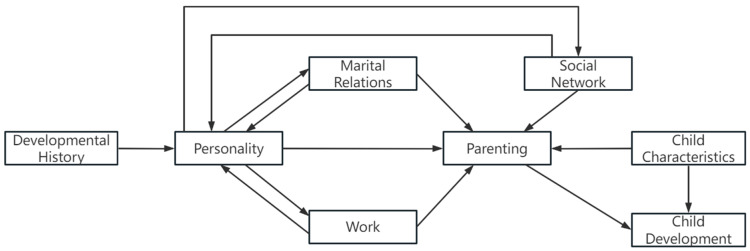
A process model of the determinants of parenting.

**Figure 2 behavsci-15-00533-f002:**
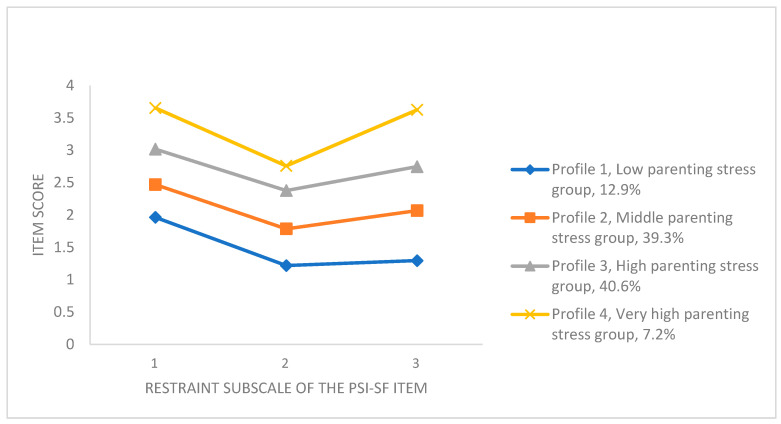
The four profiles of the best-fitting solution.

**Table 1 behavsci-15-00533-t001:** Descriptive statistics for the total sample (n = 303).

Variables		Mean ± SD/% (n)	Min–Max
Gender (males)		52.5% (159)	
Age		61.170 ± 14.830	19.000–101.000
Structure	Paternal grandparents	74.9% (227)	
	Maternal grandparents	25.1% (76)	
Family income	<2000	9.9% (30)	
	2001–3000	15.2% (46)	
	3001–4000	15.2% (46)	
	4001–5000	14.5% (44)	
	5001–10,000	27.7% (84)	
	10,000–20,000	11.9% (36)	
	>20,000	5.6% (17)	
Parenting stress		2.365 ± 0.548	1.028–3.861
Co-parenting relationship		4.989 ± 0.816	2.389–6.806
Parenting self-efficacy		3.881 ± 0.558	1.158–5.000

Family income unit: RMB.

**Table 2 behavsci-15-00533-t002:** Indexes of model fit for parenting stress solutions based on type I to V profiles.

Profiles	AIC	BIC	ABIC	Entropy	LMR-LPR (*p*)	BLRT (*p*)	Mixing Ratio
I	1781.346	1803.629	1784.600	--	--	--	--
II	1560.399	1597.536	1565.821	0.743	<0.001 ***	<0.001 ***	0.432/0.568
III	1493.555	1545.547	1501.147	0.756	0.047 *	<0.001 ***	0.257/0.175/0.568
**IV**	**1469.645**	**1536.493**	**1479.406**	**0.804**	**0.014** *	**<0.001** ***	**0.129/0.393/0.072/0.406**
V	1465.853	1547.555	1477.783	0.779	0.729	0.171	0.304/0.139/0.033/0.412/0.112

* *p* < 0.05, *** *p* < 0.001; Note: n = 303; Bold indicates the best fit.

**Table 3 behavsci-15-00533-t003:** A comparison of differences in means (standard deviations) across the three dimensions of parenting stress.

	PD	P-CDI	DC
Low parenting stress (1)	1.964 (0.675)	1.218 (0.251)	1.295 (0.248)
Middle parenting stress (2)	2.470 (0.475)	1.786 (0.378)	2.067 (0.232)
High parenting stress (3)	3.018 (0.527)	2.378 (0.391)	2.747 (0.273)
Very high parenting stress (4)	3.655 (0.658)	2.758 (0.509)	3.625 (0.258)
F	68.783 ***	136.401 ***	560.153 ***
LSD	a < b < c < d	a < b < c < d	a < b < c < d

*** *p* < 0.001; PD = Parental Distress; P-CDI = Parent–Child Dysfunctional Interaction; DC = Difficult Child; a = Low parenting stress; b = Middle parenting stress; c = High parenting stress; d = Very high parenting stress.

**Table 4 behavsci-15-00533-t004:** Multinomial logistic regression results predicting latent profiles of membership.

	**Profile 2 vs. Profile 1**			**Profile 3 vs. Profile 1**		
	**OR**	**95% CI**	**Β**	**OR**	**95% CI**	**Β**
Co-parenting relationship	0.460	0.273, 0.777	−0.776 **	0.226	0.111, 0.461	−1.488 ***
Parenting self-efficacy	0.107	0.045, 0.254	−2.239 ***	0.025	0.007, 0.084	−3.692 ***
Structure (maternal)						
Paternal	1.854	0.862, 3.990	0.617	1.094	0.371, 3.227	0.090
Income	0.821	0.652, 1.035	−0.197	0.650	0.470, 0.900	−0.431 **
Gender (female)						
Male	1.673	0.804, 3.481	0.514	4.888	1.496, 15.966	1.587 **
Age	1.005	0.981, 1.030	0.005	1.008	0.973, 1.045	0.008
	**Profile 4 vs. Profile1**			**Profile 2 vs. Profile3**		
	**OR**	**95% CI**	**Β**	**OR**	**95% CI**	**Β**
Co-parenting relationship	0.298	0.174, 0.508	−1.212 ***	2.039	1.154, 3.603	0.712 *
Parenting self-efficacy	0.034	0.013, 0.088	−3.371 ***	4.275	1.678, 10.891	1.453 **
Structure (maternal)						
Paternal	2.717	1.235, 5.978	1.000 *	1.695	0.648, 4.437	0.528
Income	0.662	0.524, 0.837	−0.412 **	1.264	0.962, 1.659	0.234
Gender (female)						
Male	1.461	0.705, 3.030	0.379	0.342	0.119, 0.988	−1.072 *
Age	1.008	0.983, 1.033	0.008	0.997	0.967, 1.028	−0.003
	**Profile 2 vs. Profile 4**			**Profile 3 vs. Profile 4**		
	**OR**	**95% CI**	**Β**	**OR**	**95% CI**	**Β**
Co-parenting relationship	1.547	1.114, 2.150	0.437 **	0.759	0.436, 1.320	−0.276
Parenting self-efficacy	3.101	1.782, 5.398	1.132 ***	0.725	0.300, 1.754	−0.321
Structure (maternal)						
Paternal	0.682	0.369, 1.260	−0.382	0.402	0.151, 1.072	−0.910
Income	1.240	1.064, 1.445	0.215 **	0.982	0.750, 1.285	−0.019
Gender (female)						
Male	1.145	0.691, 1.897	0.135	3.345	1.161, 9.635	1.208 *
Age	0.998	0.981, 1.015	−0.002	1.001	0.970, 1.032	0.001

* *p* < 0.05, ** *p* < 0.01, *** *p* < 0.001; Note: Gender: female children as the reference group; Structure: maternal grandparents as the reference group; the group behind “vs.” as the reference group for multiple regression; Profile 1: low parenting stress group; Profile 2: middle parenting stress group; Profile 3: high parenting stress group; Profile 4: very high parenting stress group; SE: standard error; OR: odds ratio.

## Data Availability

The datasets generated during and/or analysed during the current study are available from the corresponding author upon reasonable request.

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
