# Peer review of "Parenting Stress and Its Influencing Factors Among Chinese Parents in Parent–Grandparent Co-Parenting Families: A Latent Profile Analysis"

_behavsci, 2025, doi:10.3390/bs15040533_

Round 1
Reviewer 1 Report
Comments and Suggestions for Authors
Please see attachment

Author Response
General Response:
We sincerely appreciate the reviewers’ constructive and insightful comments, which have greatly helped us improve the quality and clarity of our manuscript. In response to the reviewers’ suggestions, we have carefully revised the manuscript to enhance its coherence, depth, and academic rigor.
Specifically, we have substantially revised the introduction, discussion, and conclusion sections to improve the cohesion of the text and strengthen the theoretical and empirical dialogue between our findings and prior research. We have addressed the dual nature of grandparental involvement, the sociocultural factors such as son preference and hierarchical family structures, and the nuances of co-parenting conflict and cooperation, particularly in the Chinese cultural context. These additions deepen the interpretative framework of our study and align our findings more closely with both international and Chinese scholarly discussions. Additionally, we have updated the reference list by incorporating several recent studies to enhance the timeliness and relevance of our citations. The newly added references are highlighted in red in the revised manuscript for ease of review. We have also made formatting improvements as suggested, including adjustments to the structure of headings and subheadings, and corrections to reference formatting to meet the journal’s standards. Tables have been placed adjacent to the corresponding text, and we ensured they comply with the journal’s style guidelines.
We have carefully addressed each point raised by the reviewers and detailed our responses point by point in this reply letter. We believe that these revisions have not only improved the readability and cohesion of the manuscript but also strengthened its contribution to the field. We sincerely thank the reviewers and the editorial team for their valuable time and feedback, and we hope that the revised manuscript meets the expectations for publication.
Comment 1: Title is informative, short and reflects the content and methodology used.
Response 1:
Thank you very much for your kind feedback on our manuscript title. We are glad that the title effectively reflects the study's focus and methodology. During the English editing process, we made a minor refinement to one of the punctuation marks in the title to ensure it meets academic standards and reads as smoothly as possible. The revised title is as follows: Parenting Stress and Its Influencing Factors Among Chinese Parents in Parent–Grandparent Co-Parenting Families: A Latent Profile Analysis
Comment 2: Abstract is structured and reflects and summarises the content of the manuscript. It presents the objective, the methodology used and the main conclusions.
Response 2:
Thank you very much for your positive evaluation of the abstract. We are pleased that the abstract effectively presents the objectives, methodology, and main conclusions of the manuscript. During the English editing process, we further refined the language to enhance clarity and readability. The revised abstract is as follows: Guided by family systems theory and the parenting process model, this study aimed to identify distinct profiles of parenting stress and examine their associations with parental characteristics, social–contextual factors, and child factors. A sample of 303 parents of preschool children (52.5% boys, mean child age = 61.17 months) from six urban kindergartens in southern and northern China participated in this study. Latent profile analysis (LPA) identified four distinct parenting stress profiles: the low parenting stress profile (12.9%), middle parenting stress profile (39.3%), high parenting stress profile (40.6%), and very high parenting stress profile (7.2%). Multinomial logistic regression analysis revealed that these profiles were significantly associated with parenting self-efficacy, the parent–grandparent co-parenting relationship, the co-parenting structure, family income, and the child's gender. These findings advance our understanding of the heterogeneity of parenting stress within Chinese parent–grandparent co-parenting families and offer theoretical and practical implications for future research and the development of targeted family support interventions.
Comment 3: Introduction states the relevance and importance of the study and presents the aim of the study clearly. The various subheadings help to organise the information and make it easier to understand. However, to make it easier to read and understand, as well as to improve the cohesion of the text, I suggest placing the various sub-headings of the introduction with transitional phrases and without numbering. I suggest that the title 'Parent' should instead be 'Parental Self-Efficacy', the title 'Social Context' should be 'Coparenting Relationship', hypothesis 3 should be subtitled 'Co-parenting Structure', hypothesis 4 should be subtitled 'Family Income' and before hypotheses 5 and 6 the subtitle should be 'Gender and Age of Children'. All these subheadings are under the heading 'Influential Factors of Parental Stress'. See line 361 which quotes Li (2018) et al. instead of Li et al., (2018).
Response 3:
Thank you very much for your constructive suggestions regarding the structure and cohesion of the introduction section. We have carefully revised the manuscript according to your advice: first, we have removed the numbering of the subheadings to improve readability and flow; second, we have modified the titles as recommended, changing "Parent" to "Parental Self-Efficacy," "Social Context" to "Coparenting Relationship," adding "Co-parenting Structure" as the subtitle for Hypothesis 3, "Family Income" as the subtitle for Hypothesis 4, and introducing "Gender and Age of Children" as the subtitle before Hypotheses 5 and 6. All of these subtitles are now placed under the main heading "Influential Factors of Parental Stress," as you kindly suggested. Third, we have corrected the citation format at line 324 from "Li (2018) et al." to "Li et al. (2018)" and marked this change in the revised manuscript. We sincerely appreciate your detailed feedback, which has greatly contributed to improving the clarity and academic quality of our paper.
Comment 4: Methodology is clear and well-defined, responding to the research objective. The limitations of the study are clearly presented. The statistical analysis is adequate and appropriate statistical methods were used to analyse the data.
Response 4:
Thank you very much for your positive evaluation of our methodology and data analysis. We are pleased that the clarity of our research design, the appropriateness of our statistical methods, and the transparent presentation of the study’s limitations meet your expectations. We have carefully retained these strengths in the revised manuscript to ensure methodological rigour and alignment with the research objectives.
Comment 5: The presentation of results is rigorous and appropriate, in line with the objectives set. I suggest that the tables be placed next to the respective analysis and that their format be improved. If different pages are necessary, the authors must put the first row of the table again.
Response 5:
Thank you for your valuable suggestion regarding the presentation of the results. We have repositioned the tables so that they appear next to the corresponding analysis in the text, as recommended. Additionally, we have adjusted the table formatting to comply with the journal’s guidelines, and ensured that no single table spans across multiple pages to maintain clarity and readability.
Comment 6: Discussion is relevant and well prepared.
Response 6:
Thank you for your positive evaluation of the discussion section. We are pleased that you found it relevant and well prepared. In addition to your encouraging feedback, and in response to suggestions from the other reviewer, we have further refined the discussion to enhance its depth and clarity. Specifically, we expanded the analysis to better address the dual role of grandparents, potential conflicts in paternal-grandparent arrangements, and sociocultural factors such as traditional son preference that may influence parenting stress. These improvements, made in Section 5 (Pages 12–17), aim to strengthen the interpretative depth of the manuscript and provide a more comprehensive understanding of the findings.
Comment 7: Conclusions are relevant and useful to clinical practice.
Response 7:
Thank you for your positive assessment of the conclusions. We are pleased that you found them relevant and useful to clinical practice. In addition, to further enhance the clarity and practical value of this section, we have made some refinements in response to suggestions from the other reviewer. Specifically, we have elaborated on the theoretical contributions and policy implications to ensure they are more directly linked to our empirical findings, thereby improving the applicability of the conclusions for clinical and family support practices.
Comment 8: The bibliographical references are adequate, but only 17% are from the last 5 years. The remaining 24% are between 5 and 10 years old and 57% are over 10 years old.
Response 8:
Thank you very much for your valuable observation regarding the time span of our bibliographical references. We fully acknowledge that a significant portion of the references are over 10 years old. Given the foundational nature of family systems theory and the parenting process model, we considered it essential to include these seminal works to ensure theoretical rigor. However, in response to your suggestion, we have carefully reviewed and updated our reference list by incorporating several more recent studies from the past five years to strengthen the relevance and timeliness of our citations. These newly added references, reflecting the latest research developments in this field, have been highlighted in red in the revised manuscript for your convenience. We believe that this update improves the balance of our references, and we remain committed to further integrating recent empirical findings in our future research.
Comment 9: General aspects - In general, the article falls within the scope of the journal, is useful and contains relevant information for clinical practice. The main aspect to revise is the rationale, where it would be important to improve the cohesion of the text to make it easier to read and understand.
Response 9:
Thank you very much for your positive evaluation of the overall relevance and usefulness of our manuscript. We appreciate your insightful suggestion regarding the improvement of text cohesion, particularly in the rationale section. In response, we carefully revised the introduction to enhance the logical flow between sections, especially by adding appropriate transitional phrases between the theoretical framework and the hypotheses. We also reorganized the subheadings in the introduction as per the reviewer’s earlier comments, ensuring that each section naturally leads to the next and clearly frames the research questions. Additionally, we further refined the discussion section to better integrate the empirical findings with theoretical interpretations, elaborating on the dual role of grandparental involvement and sociocultural factors such as traditional family hierarchy and son preference. These changes have improved the coherence between our findings and the broader literature. Finally, we revised the conclusion to ensure that the theoretical and practical implications are more tightly linked to the study’s results, making the take-home messages clearer for both researchers and practitioners.
We believe these improvements have substantially strengthened the clarity and cohesion of our manuscript. Thank you again for your constructive feedback, which has been very helpful in improving the quality of our work.
Special thanks to Reviewer#1 for the excellent suggestions again and sincerely hope that the correction will meet with approval.

Reviewer 2 Report
Comments and Suggestions for Authors
This study is grounded in two well-established theoretical frameworks—Family Systems Theory (FST) and the Parenting Process Model (PPM)—and the research design is constructed accordingly. The discussion effectively illustrates the interconnection between these two frameworks. A notable strength of the study lies in its application of these theories to a relatively contemporary social phenomenon, co-parenting, thereby highlighting both the relevance and limitations of classical models.
1. However, the current introduction incorporates too many subthemes, including FST, PPM, the status of co-parenting, China’s sociocultural context, and a review of previous research. As a result, the narrative structure appears overloaded. In SSCI-level journals, it is standard practice to limit the introduction to establishing the research gap, articulating the study's significance and objectives, and briefly presenting the research questions—while reserving theoretical discussions and literature reviews for dedicated sections. For instance, topics such as the prevalence of co-parenting in China, dual-income households, grandparental involvement, and demographic data could be summarized in a concise paragraph, rather than discussed at length. Given that many SSCI readers are not specialists in Chinese society, cultural context should be presented only to the extent necessary, and in a succinct manner.
2. Moreover, the introduction lacks specific comparisons with prior empirical studies on co-parenting and does not sufficiently articulate how this study contributes distinctively to the field. In particular, greater effort should be made to position the unique roles of grandparents in China within broader international discourses. Comparative perspectives—especially cross-cultural co-parenting research—should be incorporated to better highlight the study’s originality. It is also recommended to strengthen the global framing of the research problem by including references to international trends in co-parenting.
3. The study's hypotheses are not explicitly stated but are implied throughout the text. It is advisable to clearly formulate the hypotheses prior to the methodology section—for example, “Positive co-parenting relationships are expected to reduce parental stress.” Furthermore, the absence of variables related to grandparents (e.g., beliefs, caregiving styles) should be explicitly acknowledged as a design limitation.
4. The discussion would benefit from a more critical perspective. For instance, the finding that paternal grandparents are associated with higher stress levels warrants deeper sociocultural analysis. The latter part of the discussion should elaborate on inconsistencies with previous literature and address the dual nature of co-parenting—its potential to provide both support and conflict. Exploring sociocultural factors such as son preference, urban parenting contexts, and grandparents’ caregiving qualifications would enrich the paper’s depth and interpretative power.
5. Policy implications presented in the concluding section sometimes appear general and insufficiently tied to empirical findings. To enhance their credibility, it is recommended to ground these suggestions in referenced literature or comparative international examples.
6. Finally, the study relies entirely on self-reported data and does not incorporate the perspectives of grandparents—an important omission given the focus on co-parenting. Future research should consider including grandparents as respondents to obtain a more holistic understanding of the co-parenting dynamic.
Author Response
General Response:
We would like to express our sincere gratitude for the reviewer’s thoughtful and detailed comments, which have significantly contributed to the improvement of our manuscript.
First, in response to the reviewer’s suggestion regarding the language quality of the manuscript, we have utilized the MDPI English Editing Service to thoroughly polish the entire manuscript. We have attached the official editing certificate as proof of this revision (please see the attached image). We believe that the language clarity and academic expression of the paper have been greatly enhanced as a result.
Second, we have carefully addressed all the specific points raised by the reviewer throughout the manuscript. In summary:
- We extensively revised the Introduction, removing overly detailed discussions and relocating theoretical frameworks to the Literature Review section. The revised introduction now focuses clearly on the research gap, the study’s significance, and our research objectives.
- We strengthened the international framing and originality of the study by adding global trends and cross-cultural comparisons, and clearly positioned the unique characteristics of Chinese grandparental co-parenting within the broader research landscape.
- We explicitly stated all research hypotheses in a dedicated section before the methodology and revised the limitations section to acknowledge the absence of grandparent-focused variables, such as caregiving styles and beliefs, as an important design limitation.
- We enriched the Discussion section with deeper sociocultural analysis, theoretical reflection, and expanded interpretation of key findings, particularly concerning paternal grandparents, son preference, and the dual nature of grandparental involvement in caregiving.
- We improved the policy implications, ensuring they are well-grounded in our empirical results and supported by international best practices and literature.
- Finally, we expanded the Limitations and Future Directions section to emphasize the need for future research to incorporate grandparents' perspectives, explore more variables across family subsystems, and conduct cross-family type comparisons for a more comprehensive understanding.
We have marked all revisions in red in the revised manuscript for the reviewer’s convenience. We sincerely appreciate the reviewer’s valuable feedback, which has played a crucial role in enhancing the clarity, academic rigor, and practical relevance of our work.
Comment 1: The current introduction incorporates too many subthemes, including FST, PPM, the status of co-parenting, China’s sociocultural context, and a review of previous research. As a result, the narrative structure appears overloaded. In SSCI-level journals, it is standard practice to limit the introduction to establishing the research gap, articulating the study's significance and objectives, and briefly presenting the research questions-while reserving theoretical discussions and literature reviews for dedicated sections. For instance, topics such as the prevalence of co-parenting in China, dual-income households, grandparental involvement, and demographic data could be summarized in a concise paragraph, rather than discussed at length. Given that many SSCI readers are not specialists in Chinese society, cultural context should be presented only to the extent necessary, and in a succinct manner.
Response 1:
Thank you very much for your insightful comment regarding the structure of the introduction. In response to your suggestion, we have substantially revised this section to improve focus and clarity. Specifically, we removed the excessive subdivision and condensed the discussions on the prevalence of co-parenting in China, demographic trends, and sociocultural context into a concise, integrated narrative. We also relocated the theoretical discussions of Family Systems Theory (FST) and the Parenting Process Model (PPM), along with the broader literature review, to the dedicated Section 2: Literature Review, in line with SSCI journal conventions.
Furthermore, we restructured the introduction to emphasize three key components more clearly: (1) The research gap, by highlighting the limited understanding of parenting stress profiles in Chinese parent–grandparent co-parenting families, especially under Confucian hierarchical norms. (2) The significance of the study, which we have clarified from both theoretical and practical perspectives: theoretically, by extending family systems theory and the parenting process model into the specific context of multigenerational Chinese families; practically, by providing empirical evidence that can inform the development of targeted interventions — such as family counseling, communication training, and policy recommendations — aimed at alleviating parental stress and promoting healthy child development in Chinese co-parenting families. (3) The research objectives, now explicitly defined as identifying latent profiles of parenting stress and their predictors within this multigenerational context.
We believe that these improvements have made the introduction more concise, coherent, and accessible to a global readership, while also providing a clear rationale for the study. For your convenience, all revisions have been marked in red in the manuscript.
Comment 2: Moreover, the introduction lacks specific comparisons with prior empirical studies on co-parenting and does not sufficiently articulate how this study contributes distinctively to the field. In particular, greater effort should be made to position the unique roles of grandparents in China within broader international discourses. Comparative perspectives-especially cross-cultural co-parenting research-should be incorporated to better highlight the study’s originality. It is also recommended to strengthen the global framing of the research problem by including references to international trends in co-parenting.
Response 2:
Thank you very much for your constructive suggestions regarding the global framing and comparative perspectives of the introduction. In response to your valuable feedback, we have significantly enhanced this section to strengthen the study's positioning within international research discourse. Specifically, we have added a discussion of international trends in intergenerational co-parenting, including examples from the Netherlands (Geurts et al., 2015), the United States (U.S. Census Bureau, 2017), and Asian societies (Ko & Hank, 2014), to situate our study within the broader global context of increasing grandparental involvement in childcare. Furthermore, to address your point on comparative empirical perspectives, we have incorporated cross-cultural research on co-parenting outcomes. We now contrast the dual narrative of grandparental involvement observed in Western contexts—balancing its supportive and conflictual aspects (Hoang & Kirby, 2020; Kenealy, 2013; Baker et al., 2010; Barnett et al., 2012)—with the uniquely hierarchical structure of Chinese families, where Confucian filial piety shapes parent–grandparent dynamics (Hoang & Kirby, 2020; Breheny et al., 2013). This highlights the distinctive cultural backdrop of our study and its contribution to expanding co-parenting research beyond Western paradigms. Additionally, we have enriched the introduction with cross-national comparisons of parenting stress, such as the differences between China, Japan, and the U.S. (Crnic et al., 2005; Chen & Zhou, 2018; Li et al., 2017), to underscore the need for culturally grounded analyses. By doing so, we further clarify the originality and practical relevance of our study, which addresses an underexplored but increasingly important population: Chinese parent–grandparent co-parenting families.
We sincerely appreciate your insightful feedback, which has greatly helped us to refine and strengthen our manuscript. We believe these enhancements have significantly improved the clarity and academic positioning of our research. All the above revisions have been carefully marked in red in the revised manuscript for your review.
Comment 3: The study's hypotheses are not explicitly stated but are implied throughout the text. It is advisable to clearly formulate the hypotheses prior to the methodology section—for example, “Positive co-parenting relationships are expected to reduce parental stress.” Furthermore, the absence of variables related to grandparents (e.g., beliefs, caregiving styles) should be explicitly acknowledged as a design limitation.
Response 3:
Thank you very much for your constructive suggestion regarding the explicit formulation of our research hypotheses and the acknowledgment of grandparent-related variables as a design limitation. In response to your valuable feedback, we have made substantial revisions to the manuscript.
First, we have clearly stated all research hypotheses in a dedicated section preceding the methodology, as you recommended. The revised hypotheses are as follows:
Hypothesis 1: Families with higher parenting self-efficacy are more likely to be categorized in the lower parenting stress profile.
Hypothesis 2: Families with a better co-parenting relationship are more likely to be categorized in the lower parenting stress profile.
Hypothesis 3: Families in which maternal grandparents co-parent are more likely to be categorized in lower parenting stress profiles.
Hypothesis 4: Families with lower incomes are more likely to be categorized in the higher parenting stress profile.
Hypothesis 5: Families raising boys are more likely to fall into the higher parenting stress profile.
Hypothesis 6: Families with older children are more likely to fall into the lower.
We believe that this clearer articulation enhances both the coherence and scientific rigor of the manuscript.
Furthermore, in addressing your important concern regarding research limitations, we have substantially revised this section to explicitly acknowledge the omission of grandparental variables—such as beliefs, caregiving styles, and attitudes—which are essential for fully understanding the dynamics of parent–grandparent co-parenting. We have emphasized the need for future research to incorporate data from both parents and grandparents to reduce potential bias and provide a more holistic view of how intergenerational expectations influence parenting stress.
We are confident that these comprehensive revisions have effectively addressed your valuable comments and have strengthened the overall quality and robustness of our study. For your convenience, all modifications have been clearly marked in red in the revised manuscript.
Comment 4: The discussion would benefit from a more critical perspective. For instance, the finding that paternal grandparents are associated with higher stress levels warrants deeper sociocultural analysis. The latter part of the discussion should elaborate on inconsistencies with previous literature and address the dual nature of co-parenting—its potential to provide both support and conflict. Exploring sociocultural factors such as son preference, urban parenting contexts, and grandparents’ caregiving qualifications would enrich the paper’s depth and interpretative power.
Response 4:
Thank you very much for your insightful and constructive suggestions regarding the depth and critical perspective of the discussion section. In response to your valuable feedback, we have undertaken substantial revisions to enrich the discussion and provide a more nuanced interpretation of our findings. Specifically, we made the following improvements:
First, we expanded our comparison between prior research and our study findings on parenting stress profiles. Previous studies of Chinese one- and two-child families identified three stress profiles (Hong & Liu, 2020), whereas our study identified four profiles in parent–grandparent co-parenting families. We attributed this difference to the increased relational complexity in multigenerational households, which intensifies triangulated interactions and amplifies role negotiations and authority conflicts, particularly in the Chinese context (Wu et al., 2022). This framing not only strengthens theoretical interpretation but also anchors our findings in the family systems theory.
Second, we deepened the discussion of China’s unique cultural context and urban parenting environments. Specifically, we examined why paternal-grandparent arrangements are associated with higher parenting stress. We discussed the dominant role of mothers as primary caregivers (93.1% in our sample) and the cultural persistence of mother-in-law conflicts under traditional Chinese norms (Shi, 2012). Furthermore, we emphasized the critical role of communication quality in mediating intergenerational tensions, citing Hoang et al. (2019), and proposed practical strategies to enhance family harmony, including conflict resolution training, father-as-facilitator roles, and community-based workshops promoting egalitarian decision-making.
Third, we elaborated on sociocultural factors, particularly the traditional son preference in Chinese families. We noted that while boys' externalizing behaviors universally increase parenting stress (Williford et al., 2007; Barroso, 2018), the cultural emphasis on sons in China may further intensify this effect. Specifically, grandparents tend to prioritize investment in grandsons (Coall & Hertwig, 2010), which, though well-intended, can undermine parental authority and escalate household stress. We highlighted the need for grandparents to move away from patriarchal child-rearing expectations and instead adopt a supportive, facilitator role within the caregiving system.
Fourth, we enriched our exploration of the dual nature of co-parenting. We addressed the unexpected finding that child age did not significantly predict parenting stress profiles. Drawing on the work of He & Guo (2021), we proposed that grandparental involvement in caregiving likely buffers age-related parenting stress by sharing daily child-rearing responsibilities. However, we also critically acknowledged that intensive grandparental involvement can become a source of tension when compounded by hierarchical family structures and value clashes. This nuanced analysis highlights the limits of the parenting process model in multigenerational Chinese families and calls for future theoretical refinement.
We believe these substantial enhancements have significantly deepened the discussion section, providing a more critical, culturally grounded, and theoretically robust interpretation of our findings. All corresponding revisions have been clearly marked in red in the revised manuscript for your convenience.
Comment 5: Policy implications presented in the concluding section sometimes appear general and insufficiently tied to empirical findings. To enhance their credibility, it is recommended to ground these suggestions in referenced literature or comparative international examples.
Response 5:
Thank you very much for your valuable feedback regarding the policy implications in our concluding section. In response to your suggestion, we have carefully revised this part of the manuscript to ensure that the practical recommendations are more closely aligned with our empirical findings and supported by relevant literature, including comparative international examples.
Specifically, we have grounded each policy implication directly in the study’s results and enriched the discussion with references to well-established international practices: First, in light of our finding that 40.6% of parent–grandparent co-parenting families fell into the high parenting stress profile, we proposed targeted family counseling interventions aimed at alleviating conflicts arising from traditional hierarchical family structures. To support this, we drew on evidence from the Family Relationship Centre at Broadmeadows in Australia, which demonstrates how culturally sensitive family counseling can effectively mitigate intergenerational conflicts and reduce parental stress in multicultural family contexts (Ojelabi et al., 2012). We believe this example offers valuable insights for adapting similar interventions to the Chinese cultural environment. Second, our analysis revealed that families raising boys were more likely to experience higher levels of parenting stress. To address this, we recommended parenting workshops that focus on behavioral regulation and emotional coaching for parents of boys. We strengthened this recommendation by referencing cross-cultural studies, including German families (Vierhaus et al., 2013) and diverse American family groups (Williford et al., 2007), which have demonstrated the effectiveness of early interventions targeting boys' externalizing behaviors in reducing parental stress. Third, we advanced the theoretical implications of our findings by highlighting the limitations of existing models such as the parenting process model and family systems theory when applied to multigenerational Chinese families. Our results, particularly the non-significant role of child age, suggest that these frameworks should be refined to better account for cultural variables like Confucian hierarchical norms, son preference, and the caregiving roles of grandparents. We emphasized that integrating these cultural dimensions into future empirical research will not only enrich theoretical understanding but also improve the cultural sensitivity and effectiveness of policy interventions.
In summary, we have significantly improved the concreteness and relevance of our policy suggestions, ensuring they are closely tied to the empirical findings of our study and supported by international literature. All related revisions have been carefully marked in red in the revised manuscript for your convenience.
Comment 6: Finally, the study relies entirely on self-reported data and does not incorporate the perspectives of grandparents-an important omission given the focus on co-parenting. Future research should consider including grandparents as respondents to obtain a more holistic understanding of the co-parenting dynamic.
Response 6:
Thank you very much for highlighting the importance of including grandparents' perspectives in research on co-parenting dynamics. We fully agree that this is a critical aspect, especially given the central role grandparents play in parent–grandparent co-parenting families.
In response to your insightful suggestion, we have explicitly addressed this issue in the revised manuscript under the section 5.3. Limitations and Future Directions. Specifically, we acknowledged that our study relied solely on parental self-reports, and emphasized that the absence of grandparental viewpoints — including their beliefs, caregiving styles, and attitudes — limits the comprehensiveness of our findings. We have clearly stated that future research should incorporate data from both parents and grandparents to reduce potential bias and provide a more holistic understanding of how intergenerational expectations influence parenting stress.
Additionally, we expanded this section to propose further directions for future research, such as: Comparing the influence of the three subsystems (parenting subsystem, social subsystem, and early childhood subsystem) on parenting stress within parent–grandparent co-parenting families. Exploring other relevant variables within each subsystem, such as marital relationships, personality characteristics, occupational stress, friend support, and children’s temperament or behavioral problems. Considering broader cultural and historical contexts, especially given the distinct influence of Confucian family norms and the evolving urban parenting environment in China. Proposing comparative studies between parent–grandparent co-parenting families and non-grandparental co-parenting families to explore differences in stress dynamics.
We believe that these comprehensive additions to the limitations and future research section strengthen the scientific rigor of the paper and directly respond to your recommendation. All revisions have been clearly marked in red in the revised manuscript for your convenience.
Special thanks to Reviewer#2 for the excellent suggestions again and sincerely hope that the correction will meet with approval.
